# Immune Checkpoint Inhibitors in the Pre-Transplant Hepatocellular Carcinoma Setting: A Glimpse Beyond the Liver

**DOI:** 10.3390/ijms252111676

**Published:** 2024-10-30

**Authors:** Luca Marzi, Andrea Mega, Chiara Turri, Stefano Gitto, Federica Ferro, Gilbert Spizzo

**Affiliations:** 1Department of Gastroenterology, Bolzano Regional Hospital (SABES-ASDAA), 39100 Bolzano-Bozen, Italy; andrea.mega@sabes.it (A.M.); chiara.turri@sabes.it (C.T.); 2Department of Experimental and Clinical Medicine, University of Firenze, 50134 Firenze, Italy; stefano.gitto@unifi.it; 3Department of Radiology, Bolzano Regional Hospital (SABES-ASDAA), 39100 Bolzano-Bozen, Italy; federica.ferro@sabes.it; 4Department of Internal Medicine, Oncologic Day Hospital, Hospital of Bressanone (SABES-ASDAA), 39042 Bressanone-Brixen, Italy; gilbert.spizzo@gmail.com

**Keywords:** liver transplantation, hepatocellular carcinoma, immune checkpoint inhibitors, allograft rejection

## Abstract

Hepatocellular carcinoma (HCC) is the most common primary liver cancer and the third leading cause of cancer-related death worldwide. Liver transplantation (LT) is the best therapy for most patients with non-metastatic HCC. In recent years, the management of patients with HCC has considerably changed, thanks to the improvement of molecular biology knowledge and the introduction of immunotherapy. To date, systemic therapy is authorized in the Western world only in patients with advanced HCC. However, this therapy could not only stabilize the tumour disease or improve survival but could display excellent response and lead to downstaging of the tumour that finally permits LT. There are increasing reports of patients that have performed LT after pretreatment with immune checkpoint inhibitors (ICIs). However, due to the intrinsic mechanism of ICIs, graft rejection might be favoured. In addition, chronic adverse effects affecting other organs may also appear after the end of therapy. This review aims to evaluate the readiness and outcomes of LT in patients with advanced HCC who have previously undergone treatment with ICIs. It seeks to identify the challenges, risks, and benefits associated with this conversion therapy. The integration of ICIs into the treatment paradigm for advanced HCC necessitates a nuanced approach to LT. While early evidence supports the feasibility of LT following ICIs therapy, there is an urgent need for standardized guidelines and more extensive longitudinal studies to optimize patient selection, timing, and post-transplant management.

## 1. Introduction

Hepatocellular carcinoma (HCC) is the most common primary liver malignancy and the third cause of cancer death worldwide [1,2].

The choice of treatment in patients with HCC is very complex, because this neoplasm occurs in most cases in the presence of chronic liver disease and other comorbidities. In the Western world, until recently, the therapeutic strategy was indicated exclusively by the stage (or substage) of the disease according to the Barcelona Clinic Liver Cancer (BCLC) scheme [3]. Advances in the surgical and systemic therapeutic setting of HCC have further increased the complexity of managing these patients. In particular, the management of HCC is based not only on oncological staging, patient frailty and comorbidities, but also on tumour location, the multiple functional parameters of the liver and specific technical contraindications affecting treatment delivery and resource availability. At present, expert tumour committees play a central role and decision-making should be based on the innovative concept of a multi-parametric therapeutic hierarchy, where different therapeutic options are ordered according to their survival benefit (e.g., from surgery to systemic therapy) and on the inverse therapeutic hierarchy, where therapies are ordered according to their conversion or adjuvant capacity (e.g., from systemic therapy to surgery) [4].

Based on the above, liver transplantation (LT) is the therapy for patients with nonmetastatic HCC [3] which achieves the greatest survival benefit for early-stage disease (BCLC-0 and A) with 5-year survival rates of about 80% [5].

The limitations in organ transplantation for patients with HCC highlight a significant challenge in treatment protocols. The Milan criteria (MC) (single tumour < 5 cm or three nodules < 3 cm), established in 1996, have served as a benchmark for selecting patients with small, unresectable tumours, demonstrating promising recurrence-free survival rates [6].

Recent studies suggested that patients outside the MC but successful downstaging can achieve satisfying survival [6]. This finding highlights the need to update treatment guidelines to enhance access to LT for more patients, particularly those who may benefit from aggressive pre-transplant therapies to reduce tumour burden [7].

The MC, though a valuable framework for selecting liver transplant candidates with HCC, may not fully capture the complexity of tumour biology [6]. This limitation has prompted exploration of alternative criteria to better identify patients who could benefit from LT despite having larger or more numerous tumours [8].

One significant development came from Yao et al. [9] about 20 years ago, who introduced the University of California San Francisco (UCSF) criteria. These criteria (single tumour of up to 6.5 cm or up to three tumours of 4.5 cm each, with a total tumour diameter of 8 cm or less) were among the first to extend the size limitations of the MC, showing no significant difference in overall survival (OS) results compared to the more restrictive MC [9]. In response, Mazzaferro et al. [10] sought to expand these criteria, developing the “up-to-seven” (U7) criteria, also known as the new MC. This model defines eligibility by setting the sum of the largest tumour size (in cm) and the number of tumours to a maximum of seven. In their study, patients meeting the U7 criteria achieved a 5-year overall survival rate of 71.2%.

In 2016, the Toronto extended criteria were introduced [11]. Unlike the MC, these guidelines place no upper limit on tumour size or the number of lesions but exclude patients with extrahepatic metastases, venous or biliary tumour thrombus, or cancer-related symptoms such as significant weight loss (>4.54 kg) or declining performance status over three months. A key requirement for patients exceeding MC is a liver biopsy of the largest lesion to assess tumour differentiation; poor differentiation disqualifies patients from transplantation [11]. The Toronto criteria are novel in their focus on evaluating tumour biology more directly [11].

As LT for HCC has become more common, it has become increasingly important to evaluate not only tumour size but also factors like tumour biology, recipient characteristics, and donor factors that may influence survival and recurrence rates.

The Metroticket model (MM), introduced in 2013 and updated in 2018, provides a more sophisticated tool for predicting post-LT mortality risk in HCC patients. It incorporates both static factors (such as tumour size and number) and biological markers (like alpha-fetoprotein (AFP) levels). The model features staged AFP cut-offs based on tumour size and considers the number of active nodules. This approach aligns with the “two-hit hypothesis” in HCC, suggesting that tumour burden and liver function are the two primary factors influencing recurrence [11]. However, although the expansion of criteria beyond the MC allowed more patients with HCC to access LT, the MM pointed out that increased tumour burden beyond the MC could lead to worse outcomes [12].

Furthermore, a recently published study compared selection criteria for LT for HCC in terms of inclusivity and predictive ability to identify the most permissive criteria (such as Metroticket 2.0, UCSF or U7) that maintain patient outcomes. This study concluded that less restrictive criteria allow wider application of transplantation for HCC without sacrificing outcomes, with an absolute difference in 3-year OS between scores of 1.5%. Finally, all scores predicted survival with *p* < 0.001 on competing risk analysis [13].

Although recipient factors such as tumour size, number, pre-transplant AFP levels, and adherence to the MC are crucial in assessing the risk of HCC recurrence post-transplant, donor factors, including donor age, use of donation after circulatory death grafts, and ischemia time, can significantly impact transplant outcomes. These considerations are vital for optimizing patient selection and improving OS rates after LT [14].

For these reasons, the American Association for the Study of Liver Disease recommends that patients who do not initially meet the MC may still be considered for LT following successful downstaging to meet these criteria [15]. This approach has been endorsed by the United Network for Organ Sharing and serves as a pathway to convert previously ineligible patients into candidates for transplantation, ultimately expanding access and improving treatment options for those with liver cancer [15].

For a long time, the approaches used for pre-surgery downstaging were locoregional treatments i.e., ablation and transarterial therapies [15]. However, it seems evident that systemic therapy including immunotherapy may also have a fundamental role in the management of advanced HCC as a neoadjuvant treatment modality before LT.

The first described cases of the use of neoadjuvant chemotherapy in primary liver tumours date from the 1970s [16]. In that decade, Hermann described the use of adriamycin in six children with hepatoblastoma with lung metastases [16]. Three patients underwent resection of the neoplasm after reduction of the initial mass [16]. This chemotherapy regimen allowed resection of a previously unresectable hepatoblastoma and reduced the morbidity and mortality of an otherwise extensive operation [16]. The development of immune checkpoint inhibitors (ICIs) has revolutionized cancer treatment, enabling the possibility to increase long-term survival in patients with metastatic disease and providing new therapeutic indications in early stages.

Currently, ICIs are also being added as a therapeutic armamentarium for the treatment of unresectable or advanced HCC that is not amenable to curative or locoregional therapy. Two groundbreaking studies, IMbrave150 [17] and HIMALAYA [18], demonstrated a significant increase in OS and progression-free survival (PFS) in patients with unresectable HCC treated with ICI compared to those who had received sorafenib, a multikinase inhibitor, the only approved therapy for almost a decade [19].

The combination of atezolizumab and bevacizumab (IMbrave150 study) compared to sorafenib demonstrated superior response rates and OS (29.8% vs. 12% and 19.8 vs. 13.4 months, respectively; Hazard Ratio (HR) 0.66, 95%, Confidence Interval (CI) 0.52, 0.85; *p* = 0.0009) [17]. HIMALAYA trial was designed to evaluate STRIDE (Single Tremelimumab Regular Interval Durvalumab) versus sorafenib in patients with unresectable HCC who had not been previously treated with systemic therapy [18]. In this global, open-label, Phase 3 study, median OS was 16.43 months (95% CI, 14.16–19.58) with STRIDE and 13.77 months (95% CI, 12.25–16.13) with sorafenib [18]. OS at 36 months was 30.7% and 20.2%, respectively [18].

Immunotherapy may enhance the effectiveness of downstaging, potentially increasing access to LT for patients. It could help maintain disease stability while on the waiting list and reduce the risk of radiologically non-definable micro-metastases [20,21].

This review examined existing data on the use of immunotherapy for downstaging, structured according to the hierarchy of study designs in the literature. We evaluated the impact of immunotherapy on LT outcomes, as well as the potential long-term effects on patients following transplantation. By synthesizing these findings, we aim to provide a comprehensive understanding of how immunotherapy can influence both pre- and post-transplant scenarios.

## 2. Immunotolerance in the Liver

The liver, located between the splanchnic and systemic circulations, acts as an immune gatekeeper [22,23]. This organ is continuously exposed to the attack of pathogens of intestinal origin and therefore has an immunosuppressive activity that inhibits the antigenic response mediated by T cells. This response is maintained by innate immune cells such as Kupffer cells, hematopoietic stem cells, hepatic stellate cells, dendritic cells (DCs), regulatory T cells, and liver sinusoidal endothelial cells [24,25] and by immune checkpoints. These are co-inhibitory molecules expressed by effector lymphocytes to inhibit hyperactivation [26]. Although these mechanisms protect the liver from continuous insults caused by antigens, they can also promote the growth of malignant tumour cells within the liver parenchyma.

Programmed cell death protein 1 (PD-1) is a critical immune checkpoint that regulates T cell responses [27]. When PD-1 binds to its ligands, PD-L1 or PD-L2, it inhibits T cell activation, thereby reducing the immune response [27]. PD-1 is expressed on activated T cells, B cells, natural killer cells, and DCs, while PD-L1 is found on antigen-presenting cells and tumour cells, and PD-L2 is primarily on DCs and macrophages [27].

The binding of PD-1 to its ligands activates intracellular signalling pathways that promote immunosuppression and plays a role in the advanced phase mainly in peripheral tissues. Specifically, phosphorylation of the immunoreceptor tyrosine-based switch motif leads to the recruitment of Src Homology region 2 domain-containing Phosphatase-1/2, which inhibits positive signalling from T cell receptor and CD28 interactions [27]. This suppression impacts crucial pathways like RAS-MEK-ERK and Phosphoinositide 3-kinase—Protein Kinase-B—mechanistic Target of Rapamycin (PI3K-Akt-mTOR), resulting in decreased production of pro-inflammatory cytokines such as tumour necrosis factor, interferon-γ, and interleukin-2 [28]. Cytotoxic T lymphocyte-associated antigen 4 (CTLA-4) similarly inhibits T cell activation, but its effects occur primarily in the lymph nodes during the initial phases of the immune response. CTLA-4 competes with CD28 for binding to CD80/CD86, and the combination of CTLA-4 and CD80/CD86 induces downstream signal transduction through PP2A and inhibition via the PI3K-Akt pathway [29,30]. Although CTLA-4 and PD-1 have overlapping functions in immune regulation, they operate at different stages of T cell activation [31]. Therefore, simultaneous blockade of both the PD-1/PD-L1 and CTLA-4 pathways can effectively restore T cell activity, enhancing the immune response against tumours. This dual approach is particularly promising in cancer immunotherapy, as it may improve therapeutic outcomes by overcoming multiple mechanisms of immune evasion [31].

## 3. Tumour Microenvironment and Immunotherapy

Carcinogenesis-associated inflammation leads to the accumulation of various immune cells, such as T lymphocytes, macrophages, neutrophils, and DCs, in the tumour and surrounding tissues [32]. This accumulation contributes to tissue remodelling and functional impairment. Alongside these immune cells, non-immune components—including fibroblasts, endothelial cells of blood and lymph vessels, and the extracellular matrix—make up the tumour microenvironment (TME). The TME plays a crucial role in supporting tumour growth and influencing the response to therapy [33,34,35].

In the context of HCC, the TME is composed of various immune cell types, including antigen-presenting cells (APCs) specific to HCC, regulatory T cells (Tregs), natural killer T cells, myeloid-derived suppressor cells, tumour-associated macrophages, and Tumour-Infiltrating Lymphocytes [36].

Major aetiologies of HCC—such as chronic hepatitis B and C virus infections, alcoholism, metabolic dysfunction-associated steatotic liver disease, obesity, and metabolic syndrome—contribute to the complex landscape of the TME [37]. For instance, chronic hepatitis B virus infection drives non-resolving inflammation, with viral covalently closed circular DNA contributing to the activation of non-classical nuclear factor-kappa B (NF-κB) pathways. This process results in the production of cytokines like transforming growth factor beta-1 (TGF-β1), which recruit various inhibitory immune cells, including myeloid-derived suppressor cells, tumour-associated macrophages, regulatory T cells, and tumour-associated neutrophils. These cells, in turn, suppress the antiviral and anti-tumour activities of cytotoxic CD8+ T cells, natural killer cells, and DCs, creating an immunosuppressive TME that fosters HCC progression [37]. The density and diversity of tumour-infiltrating immune cells play a pivotal role in determining prognosis and predicting the efficacy of therapies in HCC. Understanding the differences in immune cell composition between primary tumours and metastatic sites within the TME is crucial for tailoring immunotherapy approaches [38,39]. Studies have shown that the immune cell landscape can vary significantly among individuals with the same cancer type, highlighting the importance of mapping the composition and functional status of immune infiltrates for both diagnosis and treatment strategy development [40,41,42,43].

Zhang et al. [34] categorized HCC into three subtypes based on the integration of single-cell and bulk data: immunodeficient, immunocompetent, and immunosuppressive.

The immunodeficient subtype of HCC, similar to “cold” tumours, is marked by reduced lymphocyte infiltration, which diminishes the effectiveness of ICIs [34]. To address this challenge, combining ICIs with therapies such as tyrosine kinase inhibitors, oncolytic viruses, or other strategies aimed at enhancing lymphocyte infiltration could be beneficial [34]. These combination approaches may help create a more favourable tumour microenvironment, potentially improving the response to immunotherapy for patients with this subtype [34]. The immunocompetent subtype of HCC, which aligns with the immune-activated subtype, demonstrates normal T cell infiltration and is linked to a favourable prognosis [34]. This subtype indicates a more active immune response, making it a promising target for therapies like ICIs, which can further enhance the anti-tumour immune activity [34]. The presence of robust T cell infiltration suggests that patients in this category may respond better to immunotherapies, potentially improving their overall outcomes [34]. Combining ICIs with T cell stimulators like IL-12 may further boost anti-tumour immunity, especially in patients with the immunosuppressive subtype of HCC [34]. This subtype, similar to the immune-exhausted subtype, features a high infiltration of immunosuppressive cells, including regulatory T cells, regulatory B cells, and macrophages, alongside the upregulation of immune checkpoints such as PD-1, PD-L1, and T cell Immunoglobulin and Mucin-domain containing-3 [34]. In this context, monotherapy with ICIs may help sustain or even reverse T cell exhaustion, potentially restoring effective anti-tumour responses [34]. Targeting both the immune checkpoints and the immunosuppressive microenvironment could enhance therapeutic outcomes for patients in this subtype [34]. This intricate interplay of immune cells within the TME not only influences tumour behaviour but also has significant implications for the development of effective immunotherapeutic strategies, emphasizing the need for personalized approaches based on the immune landscape of each patient’s tumour [34].

## 4. Use of Immunotherapy as Downstaging in the Pre-Transplant Setting

### 4.1. Case Reports

The first published case, in which immunotherapy was used in a patient with hepatitis C virus (HCV) cirrhosis and HCC who underwent a LT, was in 2019 in Tennessee. The patient died due to acute liver necrosis after receiving nivolumab at a dose of 240 mg every 2 weeks for about 2 years and transplanted 8 days after the last dose [44].

The second case report was published in the following year (2020) in which a patient with alcoholic cirrhosis and HCC began nivolumab therapy in June 2017 for a total of 24 cycles and was discontinued 6 weeks prior to inclusion on the transplant list. One year after LT, the patient had neither tumour recurrence nor rejection [45].

The considerable interest in the use of immunotherapy in advanced stage as downstaging was such that four case reports were published in 2021 [46,47,48,49]. The case reports described in that year vary from cases of complete absence of rejection [47,49] to fatal rejection with the need for re-transplantation [48] or death [33]. For example, the case treated with durvalumab monotherapy and placed on the LT list after a 3-month interruption showed no rejection events or disease recurrence [47]. Also in 2021, the case of a re-transplant was described after demonstrating severe acute rejection with massive hepatic necrosis and loss of the first allograft, attributed to immune-mediated damage [48]. Before the transplant, the patient had been treated with nivolumab (240 mg every 2 weeks in the first month, then 480 mg every 4 weeks for 15 months) and stopped 5 weeks before the transplant [48]. Following re-transplantation, therapeutic plasmapheresis, intravenous immunoglobulin, and preoperative anti-thymocyte globulin were used to reduce DSA and significantly lower the immune response [48].

In 2022, three notable case reports demonstrated promising results regarding the use of immunotherapy in managing rejection and improving outcomes in LT [50,51,52]. The first patient exhibited an excellent response to a combination therapy of atezolizumab (an anti-PD-L1 antibody) and bevacizumab (an anti-VEGF antibody) prior to LT [50]. Imaging studies revealed a significant response to the treatment, leading to successful LT performed eight weeks after the last dose of immunotherapy [50]. The second patient received three cycles of pembrolizumab (anti-PD-1) at a dose of 2 mg/kg every three weeks [51]. The LT took place 138 days after the last dose of pembrolizumab. However, the patient encountered complications shortly after transplantation, requiring a second liver transplant just six days later due to an arterial issue [51]. In the third case, the patient experienced acute cellular rejection within the first two weeks post-LT [52]. This rejection was successfully managed with high-dose glucocorticoids, followed by thymoglobulin treatment. Prior to transplantation, the patient had received nivolumab at a dosage of 480 mg every four weeks for a total of 23 cycles, with LT occurring 16 days after the last nivolumab infusion [52]. Notably, the liver explant analysis revealed no viable HCC, no signs of vascular invasion, and no extrahepatic tumour extension [52].

The second published case report with excellent complete tumour response after atezolizumab/bevacizumab and subsequent LT for liver failure was described in 2023 [53]. At 10 months after LT, no HCC recurrence or rejection occurred [54].

Only a few considerations can be deduced from these case reports. The patient who died after receiving immunotherapy had been treated with nivolumab for nearly two years and underwent LT just eight days after the final dose. In contrast, other patients who received their last dose of immunotherapy closer to the time of transplantation did not experience fatal outcomes. This suggests that the timing of the last immunotherapy dose relative to the transplant may significantly influence post-transplant outcomes. Nivolumab has a half-life of approximately four weeks, which means that the drug can remain in the body for an extended period after administration. Given this pharmacokinetic profile, it is crucial to avoid a short interval between the last dose of nivolumab and the timing of the LT. Careful planning regarding the timing of immunotherapy could help mitigate the risk of adverse events and improve transplant outcomes [44,45,46,47,48,49,50,51,52,53,54].

A brief description of all reported cases is summarized in Table 1.

### 4.2. Case Series

In 2021, three case series were published, none of which reported rejection leading to re-transplantation or patient death [55,56,57]. Tumour recurrence occurred in two patients: one developed metastases in the liver, vertebrae, and lungs after 7 months, and another experienced lung recurrence after 3 months [57]. The first case series involved nine patients treated with nivolumab (240 mg every 2 weeks) prior to LT, with eight patients (89%) receiving their final dose within 1 month of LT [55]. Mild acute rejection due to low tacrolimus levels (<6 ng/mL) was observed, which resolved with dose adjustment [55]. One patient (11%) received a transplant from a living donor [55]. In the second retrospective cohort, seven patients were treated with neoadjuvant pembrolizumab (200 mg every 3 weeks) or camrelizumab (200 mg every 2 weeks) alongside lenvatinib, followed by a 42-day washout period before LT. One patient experienced transplant rejection post-LT, which was managed by adjusting the immunosuppressive regimen [56]. Lastly, a case series analyzed five patients with HCC who received anti-PD-1 therapy before LT, with a mean washout period of 63.80 ± 18.26 days [57].

Two additional case reports were published in 2022 [58,59]. The first report described five patients who experienced liver necrosis and graft loss when the interval between the last immunotherapy dose and LT was less than 3 months [58]. The second, a case-control study involving 86 patients, included eight who received nivolumab, of which five underwent LT [59]. The median time from the last ICI dose to LT was 105 days (range: 11–354). Among the ICIs-treated patients, two (40%) had biopsy-confirmed rejection, compared to 3 patients (6.4%) in the non-ICIs group. Additionally, two graft losses occurred in the ICIs group (40%) [59]. Notably, both patients with biopsy-confirmed rejection had received ICIs therapy within 90 days before LT, while none of the three patients who had their last ICIs dose more than 90 days before LT experienced rejection [59].

Another case series published in 2023 and 2024 confirmed the data of previous years in terms of post-LT rejection [60,61,62].

In a Chinese retrospective study in which two patients received nivolumab (3 mg/kg every 2 weeks for six and four cycles, respectively), seven pembrolizumab (200 mg every 3 weeks), four sintilimab (200 mg every 3 weeks), and two camrelizumab (3 mg/kg every 3 weeks) and one nivolumab, toripalimab, sintilimab, and tislelizumab (27 cycles), no immunocorrelated reactions occurred [60]. However, tumour recurrence after surgery was described in five patients and the 1-year tumour recurrence rate was 25.0% [60]. Three case series confirmed the safety of atezolizumab/bevacizumab or ipilimumab/nivolumab before liver transplantation with efficacy on tumour and absence of organ rejection or non-healing of the wound after transplantation [61]. Finally, a case series also confirmed the above finding in nine patients receiving atezolizumab/bevacizumab, ipilimumab/nivolumab, nivolumab, or pembrolizumab [62].

A brief description of all reported cases series is summarized in Table 2.

### 4.3. Systematic Review and Meta-Analysis

A recent systematic review and meta-analysis of 91 patients with HCC treated with immune checkpoint inhibitors in the pre-liver transplant setting confirms acceptable overall post-transplant outcomes [63]. Among the 91 patients, there were 24 (26.4%) allograft rejections, nine (9.9%) HCC recurrences, and nine (9.9%) deaths. The median (IQR) washout period for patients with ≤20% probability of allograft rejection was 94 (196) days. However, OS did not differ between cases with and without allograft rejection (log-rank test, *p* = 0.2). Age and duration of ICI washout are related to the risk of allograft rejection, and a 3-month washout may reduce it to that of patients without ICI exposure. Furthermore, a higher number of ICI cycles and a tumour burden within the MC at completion of immunotherapy may predict a reduced risk of HCC recurrence, but this observation requires further validation in larger prospective studies [63]. Another published review considers immunotherapy to be appropriate to improve the efficacy of downstaging in those with a more advanced tumour burden or to maintain a more durable response while awaiting LT [10].

### 4.4. Clinical Trials

The XXL study [48] is the first prospective trial to expand the MC, demonstrating that effective and prolonged downstaging therapy significantly improves post-transplant prognosis [64]. The findings of this study have transformed the treatment paradigm for HCC, establishing a new standard of care for intermediate-stage HCC unresponsive to local therapies and for advanced-stage disease [48]. However, there is currently limited data supporting the use of systemic immunotherapy as a bridging or downstaging strategy before LT. Several ongoing studies are investigating the safety and efficacy of neoadjuvant immunotherapy in patients awaiting LT.

In the trial entitled “Atezolizumab and Bevacizumab Pre-LT for Patients with HCC Beyond MC” (NCT05185505), patients with HCC beyond MC will be treated with neoadjuvant/downstaging atezolizumab for 6 months plus bevacizumab and transarterial chemoembolization (TACE) before LT. This multi-site study will involve a site in the USA and a site in Canada. Combined enrolment from these sites envisages the recruitment of up to 30 patients. The authors hypothesize that atezolizumab and bevacizumab can be used in the pre-transplant setting without increasing the risk of rejection at 1-year post-transplant.

The PLENTY202001 study (NCT04425226) is designed to assess the safety and efficacy of pembrolizumab combined with lenvatinib in patients with HCC exceeding the MC prior to LT. The primary goal of the study is to determine whether this drug combination is superior to standard waitlist management in terms of recurrence-free survival (RFS) and objective response rate (ORR).

The Dulect2020-1 study (Durvalumab and Lenvatinib in participants with locally advanced and metastatic HCC-NCT04443322) is a prospective, open-label study involving 20 patients, designed to assess the safety and efficacy of durvalumab combined with lenvatinib in patients with advanced HCC prior to LT. The study’s primary aim is to determine whether patients with locally advanced HCC can benefit from this combination therapy before transplantation. Additionally, it will evaluate whether patients with unresectable metastatic HCC experience improvements in progression-free survival (PFS) or RFS following LT, as well as examining objective response rate (ORR) and OS.

ESR-20-21010 is a Phase II (NCT05027425), single-arm, multicentre clinical trial designed to evaluate the safety (in terms of proportion of patients experiencing rejection, within 30 days of transplant) and efficacy of STRIDE regimen for the treatment of patients with HCC before LT. Thirty patients will be enrolled, and an interim analysis will be performed after ten patients to ensure safety. Patients will be treated with the immunotherapy combination for up to 4 months. After a minimum washout period of 28 days, they will undergo locoregional therapy according to the guidelines and after a minimum washout period of 72 days, they will undergo LT.

A single-centre, prospective, non-interventional cohort study (NCT05411926) aims to enrol 60 patients awaiting LT for HCC. This study will compare two groups: 30 patients with a history of PD-1/PD-L1 monotherapy and 30 patients with no history of PD-1/PD-L1 monotherapy. Key endpoints include the incidence and timing of acute rejection, Banff classification, rejection-related mortality, cellular immune function, tacrolimus dosing and concentration, as well as OS and RFS. The study (SHR-1210-NCT04035876) aims to evaluate the primary effects and safety of camrelizumab in combination with apatinib in patients with HCC prior to LT. Participants will receive camrelizumab (200 mg intravenously every 2 weeks) and apatinib (250 mg orally once daily) in 4-week treatment cycles. Each patient will undergo at least two cycles of camrelizumab, with the drug discontinued 5 weeks before LT, while apatinib will be stopped 1 week before the procedure. The study will assess ORR, RFS, OS and time to progression (TTP).

Additionally, a study by Sun Yat-sen Memorial Hospital Organ Transplant Center (NCT05913583) will investigate the relationship between the use of immune checkpoint inhibitors (ICIs) and the incidence of post-transplant rejection, rejection-related death, or graft loss within 1 year after liver transplantation. Secondary objectives will include identifying risk factors for graft rejection and post-transplant complications, such as early allograft dysfunction, bleeding, infection, and biliary or vascular complications.

Most of these studies take place in North America and China. Their results could also have an impact on the number of patients not eligible for transplantation due to tumour burden. Ongoing studies are listed in Table 3. Available online: https://www.clinicaltrials.gov/ (accessed on 26 October 2024).

## 5. Immunosuppressive Treatment Options Post-LT

Although there is a rationale for the introduction of immunotherapy in downstaging, there are no randomized clinical trials evaluating the role of this strategy on liver transplant outcomes. Available data come from cases in which immunotherapy was administered to patients with advanced disease who were initially unsuitable for LT, and they were subsequently placed on the transplant list [65]. Tabrizian P et al. described a series of nine patients with HCC who received nivolumab therapy and successfully transplanted [55]. Fifty-six percent of patients had hepatitis B and 56% (five patients) had undergone HCC resection; one transplant (11%) was from a living donor. Nivolumab was administered at a dose of 240 mg every 2 weeks and 89% (eight) of patients received the last dose within 4 weeks of transplant [55]. ICIs used in the cases described above include anti-PD1 (camrelizumab, nivolumab, pembrolizumab, sintilimab, toripalimab), anti-PD-L1 (atezolizumab, durvalumab), and anti-CTLA-4 (ipilimumab) antibodies. The choice of the type of ICI could play a key role in this patient setting both about hepatotoxicity and duration of action. For example, CTLA-4 inhibitors have the highest rate of hepatotoxicity [64]. The incidence of hepatotoxicity varies between 0–30% with 1–20% of grade 3/4 severity [66]. The frequency appears to increase when higher doses of CTLA-4 inhibitors are administered. Studies on PD-L1 inhibitors have shown rates of hepatotoxicity like those observed for CTLA-4 inhibitors (1–17%, 3–5% grade 3/4) [67]. In contrast, monotherapy with PD-1 inhibitors has a lower incidence of hepatotoxicity (0–3%) and very few grade 3/4 reactions (<1%) [66]. The incidence of immune-related adverse events secondary to the latter two does not appear to be dose-related [66].

Pharmacological effects and the likelihood of drug interactions are influenced by the half-life of the drug. Nivolumab (anti-PD-1), pembrolizumab (anti-PD-1), and atezolizumab (anti-PD-L1) have a half-life of 27 days, durvalumab (anti-PD-L1) of 17.8 days, while ipilimumab (anti-CTLA4) has a half-life of 15 days [67]. In this regard, living donor LT (LDLT) could be useful. LDLT is a widely used strategy in Asia while reduced in the US [68]. Regarding the administration of ICIs prior to LT, LDLT could be a promising strategy because it would be possible to define the timing of drug administration according to the date of surgery. This strategy would allow ICIs and anti-VEGF therapies to be discontinued for a certain period before LT.

In consideration of the very interesting but very complex topic from a clinical and outcome point of view, the European Society of Organ Transplantation (ESOT) has convened a working group to address the current state of downstaging, bridging, and immunotherapy in liver transplantation for HCC [69]. To date, due to insufficient evidence, no conclusive recommendation can be made about the use of immunotherapy before liver transplantation [69].

## 6. TACE/TARE and ICIs for Downstaging and Bridging to Liver Transplant

Recent studies have explored the effectiveness of combining LT with immunotherapy and locoregional therapies (LRTs), aiming to improve outcomes and reduce recurrence rates. LRTs, such as transarterial embolization (TAE) and ablation, can further reduce tumour burden when used alongside ICIs [70,71,72,73,74,75,76,77,78,79,80,81,82,83,84,85,86,87,88].

Wang et al. [70] reported on the use of transarterial chemoembolization (TACE) combined with atezolizumab and bevacizumab in 2023, noting an ORR of 42.9%. While the median follow-up was limited and the sample size small, this ORR is notably higher than the 17.7% reported by Hiraoka et al. [71] for atezolizumab and bevacizumab alone in patients with BCLC B HCC.

A larger retrospective study by Cao et al. [72] further supported these findings, showing that patients receiving TACE- atezolizumab and bevacizumab experienced significantly longer PFS (10 months) and OS (14 months) compared to those receiving atezolizumab and bevacizumab alone (6 months and 9 months, respectively).

In addition, a 2023 trial by Yu et al. [73] combined transarterial radioembolization (TARE) with atezolizumab and bevacizumab, reporting impressive results: a 12-month PFS of 66.7% and an OS of 77.1%. Early data suggest that TARE, like ablation, may induce the abscopal effect and promote immunogenic cell death [74]. Studies indicate that TARE is associated with increased levels of pro-inflammatory cytokines, enhanced expression of tumour-associated antigens, and recruitment of lymphocytes, further emphasizing its potential role in immunotherapy [75].

The predictive capacity of lymphocyte and cytokine counts in assessing TARE efficacy bolsters the idea that TARE not only serves a therapeutic role but also enhances the immunological response against tumours, paving the way for more effective combination strategies in the treatment of advanced HCC [76].

In a study from Wehrle et al. [77], a combination of LRTs with either ICIs or anti-EGFR agents prior to LT resulted in a 1-year recurrence-free survival (RFS) rate of 75% (six out of eight patients), emphasizing the significance of achieving effective tumour downstaging. Additionally, a separate protocol combining TAE or ablation with ICIs showed no recurrence in all nine patients followed for an average of 16.5 months [78].

Due to the lack of data specifically addressing the combination of locoregional therapy and immunotherapy in the context of transplantation, a review of the literature from non-transplant settings was conducted.

The first systematic review included 19 studies comparing TACE or radiofrequency ablation (RFA) with immunotherapy, but it did not assess safety profiles [79]. The second review, which involved four studies comparing TACE with dendritic cell therapy, found that patients receiving the TACE combined with dendritic cell and cytokine-induced killer (TACE-DC-CIK) therapy were more likely to experience fever compared to those in the control group (*p* = 0.001) [80]. In five prospective studies (one randomized controlled trial and four non-randomized trials), no significant safety differences between treatment arms were reported [81,82,83,84,85,86].

However, the small sample sizes in these studies limited the reliability of the conclusions. Among the seven non-randomized retrospective studies, five examined early death or severe complications, with none reporting major complications or deaths related to the treatments assessed and the remaining two studies did not provide safety data [87,88].

While these findings offer some insights into the safety and potential oncologic outcomes of combining locoregional therapy with immunotherapy, they cannot be directly applied to patients undergoing liver transplantation or those on the transplant waiting list. As a result, no specific recommendations can be made regarding this combined treatment approach in the transplant setting.

## 7. Radiological and Non-Invasive Surveillance of HCC Recurrence

To effectively prevent the recurrence of liver malignancies like HCC, robust post-LT surveillance is essential. These cancers are known for their high recurrence rates, making accurate and non-invasive follow-up methods critical for patient health [89,90]. Traditionally, follow-up after LT has relied heavily on imaging techniques, such as computed tomography (CT) and magnetic resonance imaging (MRI), to monitor for metastases over several years. To optimize the use of immunotherapy in HCC patients and to assess its efficacy in a neoadjuvant setting, it is essential that tumour response can be adequately assessed and standardized. To date, there is insufficient evidence to make meaningful recommendations on how to best assess response to immunotherapy for HCC. Imaging techniques and biomarkers are needed to define tumour response, and explant analysis of specimens should be performed prospectively with careful radiology–pathology correlation. For example, in seven studies, radiological assessment was associated with objective response and significant reduction in tumour burden [17,91], without pathological assessment. In these studies, the objective response rate according to modified Response Evaluation Criteria in Solid Tumors (mRECIST) ranged from 22% to 34%, while complete response was reported in 2.2–5.5% of cases [92,93]. Three recent published studies reported both radiological and pathological response. Pathological complete response ranged from 8% to 25% and major pathological response (>70% necrosis) was observed in 20–42%, while preoperative imaging according to RECIST 1.1 reported partial and complete response in only 8–15% and 0%, respectively [94,95,96].

Given the high rate of explants exceeding the MC after transplantation, current contrast-enhanced CT and MRI techniques are insufficient in predicting treatment response [97]. Furthermore, immunotherapy-induced changes within the tumour and tumour microenvironment may influence the relationship between the degree of pathological and radiographic response [95] while vasoconstrictor and antiangiogenic effects of drugs may induce a false positive assessment of response by mRECIST [98].

However, the advent of circulating tumour DNA (ctDNA) surveillance presents a promising alternative [99,100]. Although its specificity in pre-LT settings may be limited, ctDNA has shown potential for detecting early recurrence post-transplant [90]. Recent studies indicated that ctDNA clearance was noted in five out of 10 patients who underwent sequential testing after LT for HCC, CCA, and colorectal liver metastases [100]. When utilized effectively, ctDNA testing could reduce the need for palliative treatments and enhance the quality of life for these patients [101,102]. Despite these advancements, more comprehensive studies are necessary to fully understand the role of ctDNA in surveillance, particularly since some data indicate an increase in cell-free DNA levels following procedures like TACE [37,102]. Further research will be crucial in establishing ctDNA’s reliability and improving patient outcomes in the post-transplant setting.

## 8. Liver and Extra-Liver Immune-Related Adverse Events

CTLA-4 and PD-1 are expressed on the surface of cytotoxic T cells, the ligands of which are CD80/CD86 and programmed death ligand 1 (PD-L1), respectively. The interaction of these proteins with their ligand prevents T cell activation, thus maintaining peripheral tolerance. This mechanism helps tumour cells escape cytotoxic T cell-mediated death [92]. ICIs prevent this interaction, thus displaying an anti-tumour effect [94]. The most frequent immune-related adverse events (irAEs) involve the skin, endocrine tissue, liver, colon, and lung [103].

### 8.1. Liver Immune-Related Adverse Events

Liver events (hepatitis) occur in 5–10% (1–2% grade 3) of patients treated in monotherapy and in 25–30% (15% grade 3) in patients treated with combined anti-PD-(L)1 and anti-CTLA-4 therapy [103]. Immune-related hepatitis tends to present as an asymptomatic elevation in alanine or aspartate aminotransferase with or without elevation in bilirubin [103]. The most typical morphological aspect of ICI damage is lobular hepatitis with necrosis, patchy or confluent. Liver damage caused is heterogeneous and involves lobular and periportal activity or sinusoidal histiocytosis, fibrin deposition, and central vein endotheliitis by anti PD-(L)1 and anti-CTLA-4, respectively [103]. Management of irAEs depends on the grade and includes the suspension of the drug or initiation of immunosuppressive therapy. Hepatitis usually resolves within four to six weeks after starting appropriate treatment [103]. This delayed reaction could be due to the pharmacodynamics of the drug [103]. After organ transplantation, the induction of immune tolerance is fundamental for the survival of the transplant and therefore of the patient. The PD-1 and CTLA-4 signal transduction pathways influence immune tolerance after transplantation. Studies in animal models and humans have shown that the PD-1/PDL-1 co-inhibitory pathway affects both the regulation of tolerance and transplant rejection [104]. PD-L1 expression was observed on hepatocytes, cholangiocytes, and sinusoids and PD-1 on infiltrating T cells in liver transplant recipients [105]. Furthermore, immunosuppressive agents can induce PD-1 expression, confirming the important role of this protein in transplant immune tolerance [105]. For these reasons, the use of immunotherapy has in the past been discouraged in organ transplant patients due to the risk of transplant rejection [106]. To date, however, there is considerable interest in this topic to the extent that case reports, clinical cases and ongoing studies have multiplied in just six months [107].

The recommended washout period following ICIs treatment has varied across studies, leading to discussions during the 2024 ILTS–ILCA Consensus Conference [14]. Key issues in transplant oncology methodologies were assessed, including downstaging criteria, macrovascular invasion, washout periods, recurrence care, and interactions with immunotherapy and immunosuppressive agents. The consensus recommendation established a washout period of 2–3 months [14]. This timeframe allows the immune system to recalibrate following the heightened immune response induced by ICI therapy, potentially reducing the risk of T cell-mediated rejection during LT. To refine these recommendations, adjustments in allocation policies are necessary to align with tumour responses to ICIs and the established pre-LT washout periods [14]. Furthermore, future clinical trials are essential to solidify these guidelines, as the design of clinical trials in transplant oncology presents unique complexities that differ from conventional oncology or transplant research methodologies [14]. The insights from the ILTS–ILCA subcommittee emphasize the need for tailored approaches in this evolving field [14,108].

### 8.2. Extra-Liver Immune-Related Adverse Events

The immune-mediated mechanisms through which they exert their effects determine their toxicity profiles, in particular irAEs [59]. IrAEs reported in the published studies that led to the approval of ICIs for HCC are superimposable to those observed in other disease settings. Effects are more frequent in organs exposed to multiple environmental antigens such as the skin, lungs, liver, and gastrointestinal tract or those with a greater tendency to autoimmunity such as the thyroid and joints [60]. However, irAEs can affect any organ, including the heart, bone marrow, kidneys, bones, pituitary gland, and others [109].

In KEYNOTE-224, pembrolizumab monotherapy showed a tolerable safety profile, with the most common adverse events (AEs) of any grade being hypothyroidism (n = 8, 8%) and adrenal insufficiency (n = 3, 3%) [109]. In CheckMate 040, in patients receiving nivolumab monotherapy, the most common AEs were pruritus (n = 9, 11%) and rash (n = 11, 23%) [110]. The addition of ipilimumab to nivolumab (CheckMate 040 cohort 4) was associated with a greater variety of toxicities, including rash (n = 14, 29%), pruritus (n = 22, 45%), diarrhoea (n = 12, 24%), decreased appetite (n = 6, 12%), fatigue (n = 9, 18%), adrenal insufficiency (n = 7, 14%), and hypothyroidism (n = 10, 20%) [91]. For the combination of atezolizumab with bevacizumab (IMbrave150), the most common adverse reactions were hypertension (n = 98, 29.8%), fatigue (n = 67, 20.4%), and proteinuria (n = 66, 20.1%) [17]. For the combination of tremelimumab plus durvalumab, the most common treatment-related AEs of any grade were diarrhoea (26.5%), pruritus (22.9%), and rash (22.4%) [18].

The grades of irAEs depend on the treatment regimen used. High-grade irAEs usually develop in a dose-dependent manner with regimens containing anti-CTLA4 antibodies (30–55% for the combination of ipilimumab plus nivolumab) but are not dose-dependent with anti-PD-1-PD-L1 antibodies given as monotherapies (incidence 10–15%) [111]. IrAEs usually occur during the first three months of treatment, but may occur at any time during therapy or even several months after treatment has ended [112]. Acute irAEs, although there are no robust randomized clinical trials, are managed by stopping ICIs and administering high-dose glucocorticoids (or potentially other immunosuppressive drugs for steroid-refractory irAEs) [113,114]. Although such therapy does not appear to interfere with antitumour responses, administration of steroids early in the course of therapy may result in inferior outcomes [115,116]. Although acute irAEs have so far received most of the attention due to their more dramatic clinical presentation and the need for urgent ongoing treatment, in the pre-transplant immunotherapy setting chronic irAEs might play a very important role as they might occur after transplantation. In fact, retrospective data published in 2021 suggest that chronic irAEs (defined as those persisting for >12 weeks after discontinuation of an anti-PD-1-PD-L1 antibody) occur in 43.2% of patients [117].

A group of leading international experts has developed clinical definitions for the terminology of irAEs, including terms related to the natural history of irAEs (i.e., re-emergent, chronic active, chronic inactive, delayed/late onset), response to treatment (i.e., non-steroid-responsive, steroid-dependent), and patterns (i.e., multisystem irAEs) [118]. Re-emergent irAEs occur in the same organ, at least twice, after the patient has temporarily or permanently stopped immune checkpoint inhibition and must resolve completely while the patient is not actively receiving immunotherapy [118]. Chronic irAEs persist beyond 3 months after discontinuation of the ICI and require continuous immunosuppression (e.g., colitis, inflammatory arthritis). IrAEs with delayed/late onset occur more than 3 months after discontinuation of immunotherapy [118]. In the pre-transplant immunotherapy setting, these irAEs might play a role in monitoring the post-transplant setting [118].

Although the median time for the initial onset of irAEs has been reported to vary from 2.2 to 14.8 weeks after the start of treatment [119], with more data, the possibility of new or re-emerging irAEs occurring long after the patient discontinues therapy is becoming apparent.

IrAEs occurring more than 100 days after the last dose of therapy were reported in 4% (18 of 452 patients) receiving nivolumab and in 6% (25 of 453 patients) receiving ipilimumab in CheckMate 238 [120] and more than 160 weeks after the start of treatment in three of 429 patients still under study in patients treated with pembrolizumab for melanoma in KEYNOTE-001, KEYNOTE-002, and KEYNOTE-006 [121].

However, it is not excluded that viral infection may play a role in the development of such events as in myocarditis, as well as triggering chronic autoimmune conditions, including type 1 diabetes, systemic lupus erythematosus, rheumatoid arthritis, and Sjögren’s syndrome [122,123].

A systematic review was conducted of 323 patients (from 229 studies) of which the majority (75%) had metastatic disease and primary site with melanoma (43%) and non-small cell lung cancer (31%). The most common therapies administered were pembrolizumab (24%) and nivolumab (37%). Chronic irAEs encountered were rheumatological (20%), neurological (19%), gastrointestinal (16%), and dermatological (14%). The irAEs persisted for a median (range) of 180 (84–2370) days and more than half (52%) of the patients had chronic irAEs lasting > 6 months [124]. A brief description of all reported data is summarized in Table 4.

## 9. Immunosuppressive Therapy

Management of post-LT immunosuppression varies significantly among centres. Traditionally, high-dose corticosteroids are administered upon reperfusion of the liver to broadly suppress immune responses, inducing T cell apoptosis and rapidly halting T cell proliferation. High-dose steroids are also used to manage severe irAEs.

Calcineurin inhibitors, the cornerstone of transplant immunosuppression, work by inhibiting T cell activation, proliferation, and differentiation through the impairment of interleukin-2 and other cytokine transcription. Mycophenolate acts by preferentially depleting guanosine nucleotides in T and B lymphocytes, thus inhibiting their proliferation.

There is a trend toward reducing or even eliminating steroids post-LT, which has generally yielded positive outcomes. This shift might explain some of the discrepancies in early rejection rates among patients who received ICIs before transplantation, as variations in steroid dosing can influence these rates.

Induction immunosuppression with T cell-depleting agents, such as anti-thymocyte globulin, can lead to profound T cell depletion. The use of such agents post-LT varies globally. While this approach may help prevent ICI-related rejection, lymphodepletion can stimulate surviving lymphocytes to undergo homeostatic proliferation and differentiate into memory T cells, potentially compromising graft tolerance over time [125].

Currently, there is no consensus on the optimal immunosuppressive regimen for patients undergoing LT with prior ICI therapy. Nevertheless, some studies suggest that death-censored rejection-free survival is higher in patients receiving at least one additional immunosuppressive drug alongside corticosteroids. This underscores the complexity of managing immunosuppression in the context of immunotherapy during the peri transplant period [126].

The immunosuppression protocols for patients receiving immunotherapy before LT typically involve a combination of medications aimed at minimizing rejection risk. Initial induction therapy often includes methylprednisolone, followed by maintenance with mycophenolate mofetil (MMF) and a calcineurin inhibitor, or occasionally an mTOR inhibitor. Prednisone is usually tapered over several weeks. Some protocols also incorporate additional agents like basiliximab or antithymocyte globulin (ATG) during induction, although these do not fully prevent rejection [44,45,46,47,48,49,50,51,52,53,54,55,56,57,58,59,60,61,62].

Post-LT, the management of acute rejection generally follows established guidelines, with methylprednisolone as the first-line treatment, demonstrating a high success rate in reversing rejection. For steroid-resistant cases, ATG has shown efficacy. Plasmapheresis may enhance the chances of overcoming acute rejection, potentially by clearing ICIs from the system. Adjusting baseline immunosuppression can improve outcomes; however, patient responses vary widely, and some may ultimately require new LT [44,45,46,47,48,49,50,51,52,53,54,55,56,57,58,59,60,61,62].

## 10. Conclusions

The review captures the complexities and opportunities associated with integrating ICIs in the treatment of advanced HCC, especially in the context of LT.

The potential for LT after ICI therapy is promising, particularly regarding graft safety. Understanding the immunological impact of ICIs on both the tumour and the transplanted organ is crucial. This involves evaluating risks such as graft rejection and chronic irAEs, which may not present immediately.

Addressing these challenges requires tailored immunosuppressive protocols that consider individual patient profiles. The timing of LT is critical; a careful assessment of ICI washout periods based on drug half-lives could help mitigate the risk of both tumour progression and graft rejection. Additionally, refining selection criteria for transplant candidates is essential. This should incorporate factors like liver function, tumour characteristics, response to ICIs, and overall patient health, ensuring that only those who are likely to benefit from transplantation are considered. A brief description of the complexity and variables of the treatment is described in Figure 1.

The call for standardized guidelines underscores the importance of data from ongoing trials. Collaboration among a multidisciplinary team—including oncologists, hepatologists, transplant surgeons, and immunologists—will be key to developing and implementing these protocols effectively.

Finally, the long-term impact of chronic irAEs must be carefully balanced against the potential benefits of treatment. Although incorporating ICIs into liver transplant protocols presents challenges, their potential to enhance outcomes makes this a crucial area for ongoing research.

## 11. Future Directions

Several critical areas require focused research to improve the outcomes of LT following immunotherapy in patients with advanced HCC. One important direction is the identification of predictive biomarkers that can help determine which patients are likely to respond to ICIs and benefit from transplantation. Advances in immunogenetics and personalized medicine could lead to customized immunosuppressive regimens designed to reduce the risk of graft rejection while preserving anti-tumour immunity. Additionally, the integration of novel imaging techniques and liquid biopsy technologies could allow for real-time monitoring of tumour dynamics and immune status, enabling more informed decision-making. Furthermore, investigating combination therapies that incorporate ICIs, targeted therapies, and local treatments may provide synergistic benefits, potentially enhancing survival rates and improving the quality of life for patients with advanced HCC.

## Figures and Tables

**Figure 1 ijms-25-11676-f001:**
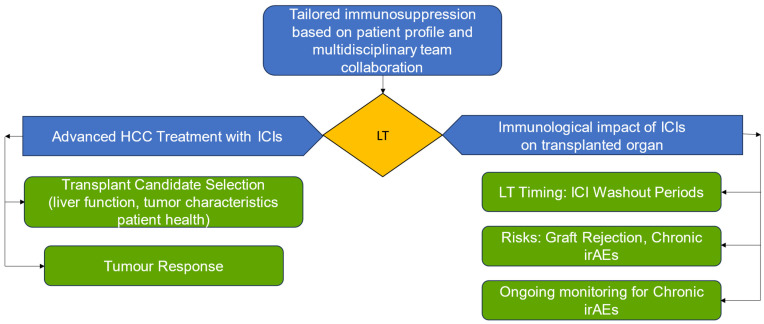
Opportunities of immune checkpoint inhibitors (ICIs) into the treatment strategy for advanced hepatocellular carcinoma (HCC) in the context of liver transplantation. Abbreviations: LT, liver transplant.

**Table 1 ijms-25-11676-t001:** Case reports published in the literature.

Study	Number of Patients Receiving ICIs Pre-Transplantation (Rejections)	Age/Sex	Under-Lying Liver Disease	Max Tumor Diameter (cm)	Max Pre-LT AFP (ng/mL)	ICI	Duration	Washout Period (Days)	Immunosoppression	Rejection Proved by Biopsy	Recurrence	Retransplantation	Postoperative Follow-Up
Nordness, 2019 [44]	1 (1)	65/M	HCV	5.5	2500	Nivolumab	24 m	8	Tacrolimus, mycophenolate mofetil, steroids	POD 6	-	No, deceased at POD 10	-
Schwacha-Eipper, 2020 [45]	1 (0)	62/M	ALD	6.4	-	Nivolumab	34 cyc	105	Unk	No rej	None	-	12 months
Chen, 2021 [46]	1 (1)	39/M	HBV	6.0	-	Toripalimab	10 m	93	Tacrolimus, steroids	POD 2	-	No, deceased at POD 3	-
Sogbe, 2021 [47]	1 (0)	61/M	HBV	4.7	1000	Durvalumab	18 m	92	Steroids, tacrolimus, mycophenolate mofetil	No rej	None	-	24 months
Dehghan, 2021 [48]	1 (1)	65/F	HCV	2.5	-	Nivolumab	15 m	35	Steroid, tacrolimus, mycophenolate mofetil	POD 10	-	Yes, POD 34	18
Lizaola-Mayo, 2021 [49]	1 (0)	63/M	NASH		1164	Nivolumab/Ipilimumab		63		No rej			
Abdelrahim, 2022 [50]	1 (0)	66/M	HCV	5	43.9	Atezolizumab/ bevacizumab	6 cyc of atezo plus 5 cyc of beva	60	Tacrolimus, mycophenolate mofetil	No rej	None	None	12 months
Kang, 2022 [51]	1 (0)	14/M	None	-	36,876	Pembrolizumab	3	138	Sirolimus, tacrolimus	No rej	None	None	96 months
Aby, 2022 [52]	1 (1)	64/M	HCV	2.4	6323	Nivolumab	23 m	16	Mycophenolate mofetil, tacrolimus, steroids	POD 9	-	No, high-dose corticosteroids	16 months
Chouik, 2023 [54]	1 (0)	57/M	ALD	6	379	Atezolizumab/ bevacizumab	18 cyc	30	Steroids, tacrolimus, mycophenolate mofetil	No rej	None	None	15 months

Abbreviations: HCV, Hepatitis C Virus; HBV, Hepatitis B Virus; ALD, Alcoholic liver disease; NASH, Non-alcohol-associated steatohepatitis; Unk, unknown; POD, postoperative day; cyc, cycles; m, months; rej, rejection.

**Table 2 ijms-25-11676-t002:** Case series published in the literature.

Study	Number of Patients Receiving ICIs Pre-Transplantation (Rejections)	Age/Sex	Under-Lying Liver Disease	Max Tumor Diameter (cm)	Max Pre-LT AFP (ng/mL)	ICI	Duration	Wash-Out Period (Days)	Immunosoppression	Rejection Proved by Biopsy	Recurrence	Retransplantation	Post-Operative Follow-Up
Tabrizian, 2021 [55]	9 (2 *)	-	HBV	2.0–21.0	3–1493	Nivolumab	-	1–253	Tacrolimus, mycophenolate mofetil, steroids	-	None	-	-
Qiao, 2021 [56]	7 (1)	53 ± 12/M	-	-	-	Pembrolizumab or camrelizumab	-	40 on average	Steroids, cyclosporine or tacrolimus, sirolimus, mycophenolate mofetil	POD 11	71% partial remission	No, corticosteroids	-
Chen, 2021 [57]	5 (0)	53.2 ± 5.4/4M, 1F	-	-	45.28 ± 33.95	Nivolumab	-	63.80 ± 18.3	Tacrolimus + mycophenolate mofetil	No	2 of 5 (POM 7, 3)	No	12 m
Schnickel, 2022 [58]	5 (2)	60/F 65/M	HCV HCV	-	-	Nivolumab Nivolumab	18 m 8 m	35 10	Tacrolimus, mycophenolate mofetil, steroids	POD 14 <POD 14	-	No, corticosteroids No, rATG rituximab or IVIGs	38 m 13 m
Dave, 2022 [59]	5 (2)	61 ± 6.5 -	- -	-	-	Nivolumab Nivolumab	- -	<90 <90	-	Yes Yes	-	Yes, successful No, death 2 months after LT	- -
Wang, 2023 [60]	16 (9)	37–67/14 M–2 F	14 HBV 2 ALD	1.5–10	2.9–38,700	2 nivolumab, 7 pembrolizumab, 4 sintilimab, 2 camrelizumab, and 1 multiple	1–27 cyc	7–184	Tacrolimus, mycophenolate or sirolimus	4 of 9 rej (POD 10, 11, 12, 30)	5 of 16 (POD 221, 108, 703, 245, 43)	No, patients returned to a normal level after adjusting the immunosuppression regimen	352.5 (median)
Ohm, 2023 [61]	3 (0)	68/M 58/M 37/M	HCV + ALD HCV HBV	2.5 2.2 3.75	3 8 2.14	Atezolizumab/ Bevacizumab, ipilimumab/nivolumab	7 cyc 4+3 cyc 6 cyc	229 2 7	Unk Unk Unk	No No No	No No No	No No No	24 m 22 m 18 m
Liu, 2024 [62]	9	-	-	-	29,523	Atezolizumab/bevacizumab, ipilimumab/nivolumab, nivolumab/pembrolizumab	-	Unk	Unk	1 of 9 rej	No	No	16.5 m

Abbreviations: HCV, Hepatitis C Virus; HBV, Hepatitis B Virus; ALD, Alcoholic liver disease; Unk, unknown; POD, postoperative day; POM, postoperative; m, months; cyc, cycles. * One attributed to low immunosuppression levels.

**Table 3 ijms-25-11676-t003:** Current and ongoing studies using ICIs before liver transplantation.

Name	Number of the Study	Study Start	Phase	Main Outcome	Expected Study Termination	Location	Recruitment Status	Etiology of Liver Disease
Atezolizumab and Bevacizumab Pre-Liver Transplantation for Patients with Hepatocellular Carcinoma Beyond Milan Criteria	NCT05185505	30.01.23	4	Proportion of patients receiving LT experiencing acute rejection	31.10.27	Houston, USA	Recruiting	HbsAg neg, HBcAb neg or HBcAb pos+ HBV DNA neg
Pembrolizumab and Lenvatinib in Participants with Hepatocellular Carcinoma (HCC) Before Liver Transplant—PLENTY 202001	NCT04425226	06.08.20	Not Applicable	Recurrence-free survival	30.12.24	China	Recruiting	No coinfection HBV-HCV
A Pilot Study of Neoadjuvant INCB099280 in Patients with Hepatocellular Carcinoma Awaiting Liver Transplant	NCT06337162	01.05.24	1	Acute Cellular Rejection Attributed to Study Therapy	01.09.28	Pennsylvania, USA	Not yet recruiting	If HBsAg+: antiviral therapy
Durvalumab (MEDI4736) and Tremelimumab for Hepatocellular Carcinoma in Patients Listed for a Liver Transplant	NCT05027425	07.12.21	2	Cellular rejection rates	07.12.30	Missouri and Ohio, USA	Recruiting	No HBsAg+ or HCV RNA+
Liver Transplantation in Patients with Partial or Complete Response After Atezolizumab Plus Bevacizumab for Intermediate-advanced Stage Hepatocellular Carcinoma: The ImmunoXXL Study–Immuno XXL	NCT05879328	23.12.22	Observational	Recurrence-free survival (RFS)	31.12.24	Milan, Italy	Recruiting	/
A Prospective, Single-arm Study of Downstaging Protocol Containing Immunotherapy for HCC Beyond the Milan Criteria Before Liver Transplantation	NCT05475613	01.08.23	2	The 2-year event-free survival rate	01.08.28	Guangdong, China	Recruiting	/
Effect of PD-1 /PD-L1 Inhibitor Therapy Before Liver Transplantation on Acute Rejection After Liver Transplantation in Patients with Hepatocellular Carcinoma	NCT05411926	17.03.21	Observational	Incidence and severity of acute rejection after LT, cellular immune function after LT, including lymphocyte subsets and cytokines, dose and drug concentration of tacrolimus after LT	09.2023	Beijing, China	Unknown status	/
Safety and Efficacy Study of Durvalumab in Combination with Lenvatinib in Participants with Locally Advanced and Metastatic Hepatocellular Carcinoma—Dulect 2020-1	NCT04443322	19.09.20	Not Applicable	Progression Free Survival Recurrence-Free Survival	31.12.25	Shanghai, China	Recruiting	No HCV + HBV coinfection
A Single Group, Open Label, Multi-center Clinical Study of Combination Camrelizumab (SHR-1210) and Apatinib for Downstaging/Bridging of Hepatocellular Cancer Before Liver Transplant	NCT04035876	2019-07-16	1-2	Objective remission rate	31.12.21	Zhejiang, China	Unknown status	No HCV + HBV

Abbreviations: HbsAg, Hepatitis B surface antigen; HBcAb, Hepatitis B core antibody; HBV-DNA, Hepatitis B Virus-DNA; LT, liver transplant.

**Table 4 ijms-25-11676-t004:** Immune-related adverse events (irAEs) in the treatment of hepatocellular carcinoma (HCC).

Aspect	Details
Mechanisms of Action	Immune-mediated mechanisms determine toxicity profiles, particularly irAEs.
Affected Organs	Commonly affected organs include skin, lungs, liver, gastrointestinal tract, thyroid, and joints. IrAEs can also affect heart, bone marrow, kidneys, and others.
KEYNOTE-224	Pembrolizumab: Hypothyroidism (8%), adrenal insufficiency (3%).
CheckMate 040	Nivolumab: Pruritus (11%), rash (23%).
Combination Therapy	Ipilimumab + Nivolumab: Rash (29%), pruritus (45%), diarrhea (24%), decreased appetite (12%), fatigue (18%), adrenal insufficiency (14%), hypothyroidism (20%).
Atezolizumab + Bevacizumab	Common AEs: Hypertension (29.8%), fatigue (20.4%), proteinuria (20.1%).
Tremelimumab + Durvalumab	Common AEs: Diarrhea (26.5%), pruritus (22.9%), rash (22.4%).
Grade of irAEs	High-grade irAEs: 30–55% for anti-CTLA4 combinations; 10–15% for anti-PD-1-PD-L1 monotherapies.
Timing of irAEs	Usually occur within the first three months but can arise anytime during or after treatment.
Management of Acute irAEs	Stop ICI treatment and administer high-dose glucocorticoids; may need other immunosuppressive drugs for steroid-refractory cases.
Chronic irAEs	Defined as lasting > 12 weeks after stopping ICI; reported in 43.2% of patients.
Clinical Definitions	Includes terms such as re-emergent, chronic active, chronic inactive, delayed/late onset, non-steroid-responsive, and steroid-dependent.
Initial Onset of irAEs	Median onset reported between 2.2 to 14.8 weeks after starting treatment.
Late-onset irAEs	Occurring >100 days after last dose: 4% with nivolumab, 6% with ipilimumab
Chronic Conditions Triggered	Possible links to viral infections and chronic autoimmune conditions (e.g., type 1 diabetes, systemic lupus erythematosus, rheumatoid arthritis).
Systematic Review Findings	In 323 patients, 75% had metastatic disease; chronic irAEs: rheumatological (20%), neurological (19%), gastrointestinal (16%), dermatological (14%).
Duration of Chronic irAEs	Median duration of chronic irAEs: 180 days (range 84–2370 days); >52% lasting > 6 months.

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
