# Peer review of "Immune Checkpoint Inhibitors in the Pre-Transplant Hepatocellular Carcinoma Setting: A Glimpse Beyond the Liver"

_ijms, 2024, doi:10.3390/ijms252111676_

Round 1
Reviewer 1 Report
Comments and Suggestions for Authors
The review is well written and concise.
I would like to commend the authors for their thorough and well-researched case study review. The detailed analysis and clear presentation of the findings demonstrate a deep understanding of the subject matter. The integration of theoretical perspectives with practical insights adds great value to the discussion. I appreciate the effort the authors have put into synthesizing relevant literature and presenting the information in a structured and engaging manner. All the references are appropriate. HCC treatment often involves liver transplantation, especially in patients with advanced disease. Understanding how to manage HCC pre-transplant is crucial for improving patient outcomes. This case study will certainly contribute to the ongoing discourse in the field. As chronic immune-related adverse events play a vital role in regarding the follow-up transplant. Can authors critically analyse what according to them could be a solution (in addition to the use of steroids) and how to balance it with immune suppression for better outcome?
However authors should try to restructure the sentences to minimize the similarity.
Sentence in Line 138-140 seems incomplete.
Comments on the Quality of English Language
Overall language is good with some very minor corrections.
Author Response
The review is well written and concise.
I would like to commend the authors for their thorough and well-researched case study review. The detailed analysis and clear presentation of the findings demonstrate a deep understanding of the subject matter. The integration of theoretical perspectives with practical insights adds great value to the discussion. I appreciate the effort the authors have put into synthesizing relevant literature and presenting the information in a structured and engaging manner. All the references are appropriate. HCC treatment often involves liver transplantation, especially in patients with advanced disease. Understanding how to manage HCC pre-transplant is crucial for improving patient outcomes. This case study will certainly contribute to the ongoing discourse in the field. As chronic immune-related adverse events play a vital role in regarding the follow-up transplant. Can authors critically analyse what according to them could be a solution (in addition to the use of steroids) and how to balance it with immune suppression for better outcome?
Thank you for your suggestion, we have critically re-analysed immunosuppressive therapies on the basis of the available clinical data, without, however, indicating a better therapy due to lack of data.
However authors should try to restructure the sentences to minimize the similarity.
We appreciate the reviewer's insightful feedback regarding sentence structure and similarity. In response, we have carefully revised the manuscript to enhance clarity and reduce redundancy. We aimed to restructure sentences to ensure a more original presentation of our findings while maintaining the integrity of the content.
We believe these revisions have improved the overall readability of the paper, and we thank the reviewer for bringing this to our attention.
Sentence in Line 138-140 seems incomplete.
The sentence has been restructured
Reviewer 2 Report
Comments and Suggestions for Authors
It is a quite interesting subject. The utilization of ICIs in the pre-LT downstaging/bridging phase is in the spotlight of scientific research
The authors successfully described a large amount of the current bibliography regarding this subject. However, I have some suggestions in order to increase the scientific value of this review.
· Moderate English editing
o For ex. (grammar/syntactic revision 52-57 line)
· The addition of a distinct paragraph regarding the adverse effects of ICIs is recommended ( & a table – including the frequency of each AE
· Add a paragraph about TME – how the etiology of chronic liver disease alters the response to ICIs ( ex. Viral hepatitis, cirrhosis MASLD , NASH-HCC with/without cirrhosis , how TME alters the response to ICIs
· Add a paragraph about TACE & ICIS for down-staging/ bridging
· The addition of a table for “ 4. Clinical evidence of the use immunotherapy: extra-liver focus” is recommended
· The paragraph of discussion could be reduced for better compression. For this reason, I recommend the complete modification of it.
o An extra paragraph regarding the immunosuppressive treatment options for post-LT phase / acute rejection & table
o The discussion paragraph must be reduced ( remove the info that you will describe in the newly added paragraphs
· You should definitely reduce the paragraph of the conclusion (up to 5 lines)
· Conclusions must be “to the point” info about your subject – the info in this paragraph are proper for the discussion
· Future direction must be before conclusions or being part of the discussion
· 50-59 reference should be modified (the reference must be the one of the published paper)
Comments on the Quality of English Languagemoderate english editing - syntactic errors
Author Response
It is a quite interesting subject. The utilization of ICIs in the pre-LT downstaging/bridging phase is in the spotlight of scientific research
The authors successfully described a large amount of the current bibliography regarding this subject. However, I have some suggestions in order to increase the scientific value of this review.
- Moderate English editing
- For ex. (grammar/syntactic revision 52-57 line)
The sentence has been restructured
- The addition of a distinct paragraph regarding the adverse effects of ICIs is recommended ( & a table – including the frequency of each AE
Thank you for the insightful comments and recommendations. In response to your suggestion, we have added a distinct paragraph discussing the adverse effects (AEs) of immune checkpoint inhibitors (ICIs) and included a table outlining the frequency of these AEs. This addition enhances the manuscript by providing a comprehensive overview of the safety profile of ICIs, which is critical for both clinicians and researchers.
- Add a paragraph about TME – how the etiology of chronic liver disease alters the response to ICIs ( ex. Viral hepatitis, cirrhosis MASLD , NASH-HCC with/without cirrhosis , how TME alters the response to ICIs
Thank you for your valuable suggestion to include a paragraph discussing the impact of the tumor microenvironment (TME) on the response to immune checkpoint inhibitors (ICIs) in the context of chronic liver disease. We recognize the importance of understanding how various underlying conditions can affect treatment outcomes.
In response to your recommendation, we will add the paragraph dedicated.
- Add a paragraph about TACE & ICIS for down-staging/ bridging
Thank you for your valuable feedback and suggestions. In response to your recommendations, we have made the following additions to the manuscript: TACE/TARE and ICIs for Down-Staging and Bridging to Liver Transplant.
- The addition of a table for “ 4. Clinical evidence of the use immunotherapy: extra-liver focus” is recommended
- The paragraph of discussion could be reduced for better compression. For this reason, I recommend the complete modification of it.
- An extra paragraph regarding the immunosuppressive treatment options for post-LT phase / acute rejection & table
- The discussion paragraph must be reduced ( remove the info that you will describe in the newly added paragraphs
- You should definitely reduce the paragraph of the conclusion (up to 5 lines)
- Conclusions must be “to the point” info about your subject – the info in this paragraph are proper for the discussion
Thank you for your constructive feedback. We agree with your suggestions and have implemented the following modifications to improve the clarity and structure of the manuscript:
- Reduction and Revision of the Discussion Paragraph: We have significantly reduced the discussion section, condensing the content to focus on the most critical points. Information that is now covered in the newly added paragraphs has been removed to avoid redundancy. The revised discussion is more concise, emphasizing key findings and their implications while omitting details now found elsewhere in the manuscript.
- New Paragraph on Immunosuppressive Treatment Options Post-LT: We have added a new paragraph discussing the immunosuppressive treatment options available during the post-LT phase, particularly in managing acute rejection. This section highlights the role of standard therapies such as calcineurin inhibitors (e.g., tacrolimus, cyclosporine), corticosteroids, and antimetabolites (e.g., mycophenolate mofetil), as well as newer agents like mTOR inhibitors.
- Future direction must be before conclusions or being part of the discussion
Thank you for your valuable feedback. While we understand your perspective on the placement of the "Future Directions" section, we believe that maintaining it as a distinct part of the conclusions provides a clearer and more cohesive closure to the manuscript. Placing future directions in the conclusions allows for a direct linkage between the key findings and the next steps in research and clinical applications, offering the reader a forward-looking perspective as they conclude the paper.
However, we are open to modifying the structure to ensure that the manuscript aligns with best practices and your expectations. As an alternative, if preferred, we could integrate future directions into the discussion section or as a separate paragraph preceding the conclusion, while keeping the conclusions more focused and succinct, as per your suggestion.
We hope this explanation clarifies our reasoning, but we remain flexible in addressing your preferences.
- 50-59 reference should be modified (the reference must be the one of the published paper)
Thank you for your careful review of the manuscript and for bringing this to our attention. We will ensure that reference 50-59 is modified to cite the appropriate published paper.
Reviewer 3 Report
Comments and Suggestions for Authors
I read with extreme interest this work. The authors summarizing information about immunotherapy pre-transplant for HCC. However, there are some things that need to be addressed.
1. The writing is very difficult to read. This is primarily because many paragraphs are as short a single sentence which is quite difficult to understand.
2. The authors comment in the introduction on AASLD recommendations related to Milan criteria. however recent data has suggested that Milan criteria are overly restrictive (PMID 31243778, 28989060, 38831488, 38241354). I understand this is not the primary focus but to me it is absolutely critical to acknowledge this expansion issue and discuss even if briefly. While the XXL study is mentioned there are many studies about extending milan that are not included.
3. The organization of the paper does confuse me. There are two introduction sections prior to section 3, which then begins clinical evidence, then there is a methods section within. Either it needs to be structured as a usual paper (Intro, Methods, Results, Discussion) or as a review, with (1. Intro, 2. Immunotolerance, 3. Clinical evidence etc.) without a methods section buried in.
4. I recommend the authors include information about circulating tumor DNA in this context. multiple studies have reported use of ctDNA to guide ICIs, including ICI+LRT in HCC which would be highly relevant for a study discussing pre-transplant therapy.
5. The section titled extra-liver focus needs to be re-framed as complications and toxicity. Then within that section should be 1. immunologic implications and 2. extrahepatic side effects.
6. The discussion is too long and is repetitive of things that have been discussed already. The authors keep adding things, even beginning multiple paragraphs with "Finally...." but then having more to follow. This is evidence of a "run on discussion". The conclusion section is also much too long. Both sections need to be condensed to highlight only the main points of the previous reviews.
7.Could the authors identify a time period where ICIs need to be stopped pretransplant? They mention the drug half lives but do not actually give a time frame needed to stop
Comments on the Quality of English Language
The english language is hard to read and needs to be improved.
Author Response
I read with extreme interest this work. The authors summarizing information about immunotherapy pre-transplant for HCC. However, there are some things that need to be addressed.
- The writing is very difficult to read. This is primarily because many paragraphs are as short a single sentence which is quite difficult to understand.
Thank you for your feedback regarding the writing style and paragraph structure. We understand your concern about the readability of the manuscript, particularly due to the use of short, single-sentence paragraphs. In response, we will revise the manuscript to ensure that paragraphs are more cohesive and flow smoothly, combining related ideas into well-developed, multi-sentence paragraphs. This will improve the clarity of the text and make it easier to follow the arguments and discussions presented.
- The authors comment in the introduction on AASLD recommendations related to Milan criteria. however recent data has suggested that Milan criteria are overly restrictive (PMID 31243778, 28989060, 38831488, 38241354). I understand this is not the primary focus but to me it is absolutely critical to acknowledge this expansion issue and discuss even if briefly. While the XXL study is mentioned there are many studies about extending milan that are not included.
We found this indication very interesting and therefore introduced the ‘up-to-7’ (U7) criteria proposed by Mazzaferro, the so-called ‘Metroticket model’ of Mazzaferro, the extended Toronto criteria, which do not stipulate an upper limit for size and number of lesions, but exclude patients with extrahepatic metastases, evidence of venous or biliary thrombi from the tumour or cancer-related symptoms, and the concept of tumour biology
- The organization of the paper does confuse me. There are two introduction sections prior to section 3, which then begins clinical evidence, then there is a methods section within. Either it needs to be structured as a usual paper (Intro, Methods, Results, Discussion) or as a review, with (1. Intro, 2. Immunotolerance, 3. Clinical evidence etc.) without a methods section buried in.
The entire article has been edited for greater understanding.
- I recommend the authors include information about circulating tumor DNA in this context. multiple studies have reported use of ctDNA to guide ICIs, including ICI+LRT in HCC which would be highly relevant for a study discussing pre-transplant therapy.
Thank you for your valuable suggestion. We appreciate your recommendation to include information on circulating tumor DNA (ctDNA) in the context of guiding immune checkpoint inhibitors (ICIs) and locoregional therapies (LRT) for hepatocellular carcinoma (HCC), particularly in the pre-transplant setting.
In response, we will add a section discussing the role of ctDNA as a biomarker for monitoring tumor burden, detecting minimal residual disease, and predicting treatment response to ICIs. Several studies have demonstrated that ctDNA can provide real-time insights into tumor dynamics, helping to tailor ICI-based therapies and evaluate their effectiveness when combined with LRT, such as transarterial chemoembolization (TACE) or radiofrequency ablation (RFA). This is particularly relevant in patients being considered for liver transplantation, where ctDNA could potentially guide decisions on down-staging therapies or bridging strategies.
- The section titled extra-liver focus needs to be re-framed as complications and toxicity. Then within that section should be 1. immunologic implications and 2. extrahepatic side effects.
Thank you for your constructive feedback regarding the organization of the manuscript. We appreciate your suggestions for improving the clarity and structure of the section titled "Extra-Liver Focus."
In response to your recommendation, we will eliminate the previous paragraph as advised and reframe the "Extra-Liver Focus" section into two distinct sub-sections: 1. Liver Immune-Related Adverse Events (irAEs) and 2. Extra-Liver Immune-Related Adverse Events.
Revised Structure:
- Liver Immune-Related Adverse Events (irAEs):
This sub-section will focus specifically on the immune-related adverse events that occur within the liver. - Extra-Liver Immune-Related Adverse Events:
In this sub-section, we will cover the immune-related adverse events that manifest in organs outside the liver. This will include discussions on dermatologic, gastrointestinal, endocrine, pulmonary, and other systemic irAEs. We will provide information on the frequency, severity, and management of these extrahepatic complications.
We believe this approach will provide a clearer and more organized presentation of the information related to immune-related adverse events associated with ICIs.
- The discussion is too long and is repetitive of things that have been discussed already. The authors keep adding things, even beginning multiple paragraphs with "Finally...." but then having more to follow. This is evidence of a "run on discussion". The conclusion section is also much too long. Both sections need to be condensed to highlight only the main points of the previous reviews.
Thank you for your constructive feedback. We agree with your suggestions and have implemented the following modifications to improve the clarity and structure of the manuscript:
- Reduction and Revision of the Discussion Paragraph: We have significantly reduced the discussion section, condensing the content to focus on the most critical points. Information that is now covered in the newly added paragraphs has been removed to avoid redundancy. The revised discussion is more concise, emphasizing key findings and their implications while omitting details now found elsewhere in the manuscript.
7.Could the authors identify a time period where ICIs need to be stopped pretransplant? They mention the drug half lives but do not actually give a time frame needed to stop
Thank you for your insightful feedback. We appreciate your suggestion to clarify the time period for discontinuing immune checkpoint inhibitors (ICIs) prior to liver transplantation, as this is an important clinical consideration.
In response, we will add a more specific time frame for stopping ICIs before transplantation, taking into account the drug half-lives and the potential risk of immune-mediated rejection. Based on current literature and clinical guidelines, ICIs are typically stopped 4 to 6 weeks prior to liver transplantation to allow sufficient clearance of the drug and minimize the risk of graft rejection. This recommendation is derived from the pharmacokinetics of ICIs, which have half-lives of 2 to 4 weeks, and the need to balance immunosuppression post-transplant with the lingering effects of ICIs.
Round 2
Reviewer 3 Report
Comments and Suggestions for Authors
thank you. the authors have addressed my concerns.
Only one suggestion, would be to add citations in the section regarding metroticket and Milan specifically citations that compare metroticket, Milan, UCSF, U7 and other criteria to each other demonstrating no difference in outcomes. This would strengthen the work. This was recently published in Transplantation and other works also.
Otherwise this is now suitable for publication in my opinion.
Author Response
Only one suggestion, would be to add citations in the section regarding metroticket and Milan specifically citations that compare metroticket, Milan, UCSF, U7 and other criteria to each other demonstrating no difference in outcomes. This would strengthen the work. This was recently published in Transplantation and other works also.
Thank you for your valuable suggestion. I agree that adding citations comparing the Metroticket, Milan, UCSF, Up-to-Seven (U7), and other criteria will enhance the depth of the discussion. Recent study has shown comparable outcomes among these criteria in terms of survival and recurrence rates. We will incorporate relevant reference to highlight that while these models differ in the factors they consider (e.g., tumor size, number, AFP levels), their overall ability to predict post-transplant outcomes is similar. This addition will strengthen the argument by providing evidence from recent literature. In addition, we have amended the paragraph on down staging criteria, making it even clearer.